# Understanding the drift of Shackleton's *Endurance* during its last days before it sank in November 1915 using meteorological reanalysis data

Marc de Vos[1], Panagiotis Kountouris[2], Lasse Rabenstein[2], John Shears[3], Mira Suhrhoff[2], Christian Katlein[4]

[1]Marine Research Unit, South African Weather Service, Cape Town, 7525, South Africa
[2]Drift and Noise Polar Services, Bremen, 28195, Germany
[3]Shears Polar Ltd, Cambridge, PE28 3LR, United Kingdom
[4]Alfred-Wegener-Institut Helmholtz-Zentrum für Polar und Meeresforschung, Bremerhaven, 27570, Germany

*Correspondence to*: Marc de Vos (marc.devos@alumni.uct.ac.za)

**Abstract.** On 5 December 1914, Sir Ernest Shackleton and his crew set sail from South Georgia aboard the wooden barquentine
vessel *Endurance*, beginning the Imperial Trans-Antarctic Expedition to cross the Antarctic continent. However, Shackleton and his crew never reached land because the vessel became beset in the sea ice of the Weddell Sea in January 1915. Endurance then drifted in the pack for eleven months, was crushed by the ice, and sank on 21 November 1915. Over many years, various predictions were made about the location of the wreck. These were based largely on navigational fixes taken by Captain Frank Worsley, the navigator of the Endurance, three days prior to, and one day after the sinking of Endurance. On 5 March 2022,
the Endurance22 expedition located the wreck some 9.4 km southeast of Worsley's estimated sinking position. In this paper, we describe the use of meteorological reanalysis data to reconstruct the likely ice drift trajectory of Endurance for the period between Worsley's final two fixes, at some point along which the vessel sank. Reconstructions are sensitive to choices of wind factor and turning angle but allow an envelope of possible scenarios to be developed. A likely scenario yields a simulated sinking location some 3.5 km from the position at which the wreck finally was found, with a trajectory describing an excursion
to the south-east and an anticlockwise turn to the north-west prior to sinking. Despite numerous sources of uncertainty, these results show the potential for such methods in marine archaeology.

## 1 Introduction

### 1.1 The Imperial Trans-Antarctic Expedition

The story of Sir Ernest Shackleton and the Imperial Trans-Antarctic Expedition has captivated historians and the public for
more than 100 years. The Expedition intended to cross the Antarctic continent, landing from the south-east Weddell Sea and

marching to the eastern part of the Ross Sea via the South Pole (Shackleton, 1919). This objective was never achieved, with Shackleton's vessel, *Endurance*, becoming beset in the sea ice of the Weddell Sea on 18 January 1915, enroute to the continental landing site. After drifting aboard the beset Endurance, having planned to wait until it broke free, Shackleton ordered the vessel abandoned in late October of 1915 due to severe damage inflicted by the crushing sea ice (Shackleton, 1919). Then, having attempted to march westward toward the islands of the Antarctic Peninsula in search of supplies and shelter, the crew was halted just a short distance from the stricken Endurance by the challenging ice conditions. There, approximately 2.5-3 km from the wreck, they established *Ocean Camp*, where they would await an improvement in conditions. After drifting with the sea ice for 10 months, and 25 days after being abandoned by the crew, Endurance finally sank during the late afternoon of 21 November 1915. Shackleton initiated a second march in late December 1915 but was again foiled by the ice conditions. Thus, *Patience Camp* was established just a week later, some 12 km from Ocean Camp, where the crew remained until early April 1916 (Shackleton, 1919). Following the break-up of the floe on which they were camping, the crew launched Endurance's three lifeboats on 9 April, sailing to and making landfall on Elephant Island on 15 April 1916. After 9 days on Elephant Island, Shackleton and five crew sailed the James Caird lifeboat to South Georgia to summon help. Thanks to some remarkable navigation from Frank Worsley, the group made landfall on southern South Georgia on 10 May (Shackleton, 1919). Shackleton, Tom Crean and Frank Worsley then crossed the Island's mountainous interior, reaching the whaling station at Stromnes on 20 May (Shackleton, 1919). The three men who had remained on South Georgia's southern shore were rescued on 21 May and after several attempts, the 22 men who remained on Elephant Island were ultimately rescued on 30 August 1916 (Shackleton, 1919). All who had set out on the Expedition survived. The Trans-Antarctic Expedition is well-documented, owing to various carefully written accounts produced by Shackleton and the crew.

## 1.2 The Search for Endurance

Despite being a point of conjecture for decades, the precise location of the wreck of the *Endurance* was unknown until 5 March 2022, when the Endurance22 expedition located it at the bottom of the Weddell Sea. From the early 2000s, several plans were drawn up to find the Endurance, with one of these coming to fruition in 2019. The *Weddell Sea Expedition 2019* was a dual-mandate scientific and archaeological undertaking (Shears et al., 2020). Though unsuccessful in finding the wreck, this expedition laid the foundation for the Endurance22 expedition (Gilbert, 2021), which began in February 2022 with much of the planning and operational team maintained.

Endurance22 was an interdisciplinary maritime archaeological project aimed at locating and surveying the wreck of Endurance. It utilised the South African research icebreaker S.A. *Agulhas II* and Saab Sabretooth autonomous underwater vehicles (AUVs) to scan a predetermined search area of the seabed using Edgetech 2105 side-scan sonar, at frequencies of 75, 230 or 410 kHz (Gilbert, 2021). A principal difference between Endurance22 and the Weddell Sea Expedition 2019 was the deployment of the AUVs in tethered mode (Gilbert, 2021). Maintaining a direct link with the vehicle and minimised the risk of communication loss, as had occurred with an AUV in the 2019 expedition (Shears et al., 2020; Dowdeswell et al., 2020). Figure 1 shows the geographical context of this study. Typical maximum and minimum sea ice extents, which occur at the end of winter and

spring respectively, are also shown. The search area and strategy were developed by marine archaeologists, historians and a
specialist subsea team who consulted archives and crew diaries (Bound, pers.comm, 14 May, 2022). To assist the wreck search, the Endurance22 expedition team also comprised sea ice researchers and meteorological-oceanographic (met-ocean) specialists to support tactical ice navigation enroute to and within the search area. Specifically, predictions of short-term ice drift direction and speed were required to assist precise subsea survey operations at depths of 3000 m, beneath completely closed drifting sea ice cover. This necessitated the use of a wide range of data sources including remote sensing data, numerical models and direct
measurements. In particular, analysis of the ice pack and the timing and magnitude of wind and tidal shifts were important in guiding the safe navigation of the vessel and the precise deployment of the AUVs. Ultimately, sea ice conditions, though challenging, were more operationally favourable than those encountered during the Weddell Sea Expedition 2019 (Rabenstein, 2022).

This aim of this study is to analyse the unknown sea ice drift between Worsley's celestial fixes on 18 and 22 November 1915.
Further, it aims to reconstruct this unknown portion of *Endurance's* last days of drift using twentieth century meteorological reanalysis data and historical weather observations.

## 2 Data and methods

### 2.1 Navigational fixes

Throughout the voyage, Captain Frank Worsley made estimates of position based on celestial sightings to track the movement
of *Endurance* through the ice pack. Endurance sank just before 17h00 local time on 21 November 1915. The definition of "local time" is nuanced, but for this study may be considered approximately similar to UTC-3. Variations in the relationship between local time and UTC are negligible given the temporal resolution of the input data and simulations used in this study. For a comprehensive explanation of the derivation of local time and uncertainties thereof, the reader is referred to Bergman and Stuart (2018a, b). Bad weather around the time of the sinking only allowed for navigational fixes three days before, and
nearly a full day after the sinking on 18 and 22 November 1915 respectively (Dowdeswell et al., 2020). The ship's exact trajectory during the intervening approximately 4 days – referred to hereafter as the *target period* – remains unknown. However, Worsley retrospectively estimated the position of Ocean Camp on 21 November, assuming it to be offset by about 1.2 nautical miles to the south-east of the 22 November position due to sea ice drift (Bergman and Stuart, 2018b). We believe this estimate was based on local wind observations, as Worsley had no means by which to observe the sea ice drift directly.
Dowdeswell et al. (2020) record that there are small uncertainties in the positions of Ocean Camp and the Endurance due to factors including: the fact that Captain Worsley made no astronomical observations between 3 days before and nearly 16 h after the sinking because of bad weather; the drift of the chronometer used (primarily affecting longitude); the exact distance and bearing between Ocean Camp (from where Worsley took a fix) and the *Endurance* (whose position he estimated by offsetting his Ocean Camp fix*)*; and the speed and bearing of the ice drift assumed for dead reckoning of the position. In this
work, we assume Worsley's fixes to be accurate, and concentrate our analyses on uncertainties introduced by the ice drift.

## 2.2 ERA-20C reanalysis data

The ERA-20C (Poli et al., 2016) is a global reanalysis produced by the European Centre for Medium-Range Weather Forecasts (ECMWF). It provides a range of atmospheric and surface ocean variables with regular spatio-temporal resolution for the period 1900-2010. Spatial resolution is approximately 125 km on the native ERA-20C triangular grid (Poli et al., 2016). However, interpolated data were downloaded on a regular grid with a resolution of 0.125° (approximately 13.9 km). The interpolated product is produced by ECMWF's Meteorological Interpolation and Regridding (MIR) package (Maciel et al., 2017) and is available via ECMWF's download portal at: https://apps.ecmwf.int/datasets/data/era20c-daily/levtype=sfc/type=an/. Temporal resolution is 3-hourly. Data are produced by a modified version of an operational atmospheric general circulation model (AGCM) and a data assimilation scheme, which form the foundation of ECMWF's Integrated Forecast System (IFS). The IFS is normally used to produce short- and medium-term weather forecasts. Modifications to the AGCM configuration and details regarding boundary conditions and forcing have been described in detail by Hersbach et al. (2015), who showed that the model successfully reproduced low frequency variability of large-scale atmospheric features. The purpose of data assimilation during production of the reanalysis is to enhance the performance of the model in simulating weather events. The meteorological observations of Hussey (see Section 2.3) have not been assimilated into the ERA-20C (Poli et al., 2016) reanalysis dataset. As such, both datasets provide independent estimates of the actual synoptic situation during the time of *Endurance's* sinking. While the ERA-20C dataset comes with large uncertainties, it has been shown to be capable of describing the large-scale atmospheric circulation and by extension, should be able to describe the wind patterns in the western Weddell Sea. We extracted 10 m wind speeds and directions from the ERA-20C dataset (Poli et al., 2016), adjusted them to the 2 m vertical level by applying a logarithmic profile correction (Manwell et al., 2009), and used them as a proxy to reconstruct the ice drift trajectory according to the methodology in Section 2.4. ERA-20C winds for the target period (along with mean sea level pressure) are shown in Figure A1 (Appendix A1). For comparability, the 2 m level was selected as a best guess for the level at which Hussey's recordings were made (see Section 2.3), as well as a representative wind condition as experienced by the sea ice floes.

## 2.3 Meteorological observations

To derive a further, independent estimate of ice drift during the target period, we requested scans of the original log of the meteorological recordings and measurements made by the expedition meteorologist, Leonard Hussey, which are kept in the Archives of the Scott Polar Research Institute, University of Cambridge. Hussey recorded surface meteorological variables generally four times per day at 12h00, 16h00, 20h00 and 00h00 GMT. Among others, wind speed and direction were measured using an anemometer and reported in units of the Beaufort wind scale, and in cardinal and inter-cardinal directions, respectively. These data were linearly interpolated to 3-hourly resolution to match the ERA-20C data (see Section 2.2) and then utilised to produce a drift trajectory for the target period in the same way as for the ERA-20C data. It should be noted that

no observations were taken during local night hours, leaving significant data gaps and introducing large uncertainties in reconstructed ice drift trajectories.

## 2.4 Reconstructing ice drift trajectories

### 2.4.1 Description of sea ice drift

To construct the historical ice drift trajectories from both datasets, we assumed a free drift regime, where sea ice motion is purely described by wind forcing and internal dynamic forces and ocean forcing are neglected. This assumption has been shown to be reasonable over short time scales for the Antarctic (Holland and Kwok, 2012; Kottmeier et al., 1992; Kwok et al., 2017; Vihma et al., 1996; Martinson, Douglas G. Wamser, 1990), since wind is the primary forcing for sea ice drift in the

Weddell Sea (Uotila et al., 2000; Vihma and Launiainen, 1993; Vihma et al., 1996). It should be noted that caution is required when applying this assumption in the Arctic, where internal ice stress, Coriolis force (due to generally thicker ice) and geographical constraints are likely to exert more control on the drift of sea ice (Lepparanta, 2011; Martinson, Douglas G. Wamser, 1990). Notwithstanding, free drift has been shown to be applicable in certain Arctic cases (e.g., Cole et al., 2014; Park and Stewart, 2016). The assumption may also break down near the coast or in mostly open water, where internal ice stress

and ocean currents respectively reduce the dependence on wind drift (Uotila, 2001). Further, free drift parameters; namely sea ice drift speed as a proportion of wind speed (hereafter wind factor; Nakayama et al., 2012) and the angle between the wind and sea ice drift vectors (hereafter turning angle; Doble and Wadhams, 2006); vary widely, even within similar time and places (Kottmeier et al., 1992) and are an important control on the drift of sea ice. This variability is reflected in the empirical derivations of wind factors and turnings angles in the literature. Recently, Womack et al. (2022) determined wind factors

ranging from 1-6% (mean 2.73%) and tuning angles ranging from -50 to 50° (mean -19.83°) for an area of the Antarctic marginal ice zone east of the study domain. In the Weddell Sea, a vast range of parameter values is reported, with wind factors of 1.5-3.5% (e.g., Kottmeier and Sellmann, 1996; Kottmeier et al., 1992; Vihma and Launiainen, 1993; Uotila et al., 2000; Martinson, Douglas G. Wamser, 1990) and turning angles of -20 to 60° (Uotila et al., 2000; Womack et al., 2022). Reported mean values are typically 2-3% and -20 to -30°, with an acknowledgement of the spread and scattering of data points.

For in-depth discussions of the free-drift assumption and its parameters, which is beyond the scope of this study, the reader is referred to the literature cited in this section. Insofar as free-drift parameter value selection affects our results, our strategy is to apply the free-drift solution to our problem using a range of realistic parameter values. In summary, we present three selected cases:

Case 1, using parameter values which both minimise trajectory error and are well within realistic ranges.

Case 2, using parameter values required to force the simulated sinking site to coincide with the actual sinking site.

Case 3, using parameters with values more typical for the Weddell Sea.

Following Womack et al. (2022) and Nakayama et al. (2012), since ocean forcing and internal ice stresses are neglected, optimised wind factors and turning angles may differ from their real values due to their implicit inclusion of these effects.

Whilst a likely scenario is identified, inferences about the unknown drift are drawn acknowledging the range of possible outcomes within the envelop produced by the different configurations.

### 2.4.2 Implementation

For each 3-hourly time step, the future position of the virtual sea ice floe is predicted by applying the wind-driven drift distance and direction to the Vincenty formula (Vincenty, 1975), as implemented in MATLAB by (Pawlowicz, 2020, last access: 31 November 2022). Figure 2 shows the resulting trajectories. A series of simulations using different wind factors and turning angles were performed (Figure A2). The effects of changing wind factors and turning angles on the resulting distance between the simulated and actual sinking sites can be seen by comparing corresponding trajectories in Figure 2 and Figure A3-Figure A4 (Appendix A1). These results guided the selection of cases described in Section 2.4.1.

### 2.5 Trajectory alignment and nudging

None of the reconstructed trajectories is able to link Worsley's fix on 18 November to his 22 November fix. While this could be due to errors in Worsley's navigation, we assume that it is mainly caused by errors in the wind forcing datasets. To overcome this limitation, we provide two additional versions of a corrected trajectory in addition to the default. For each of Cases 1-3, we therefore provide three possible trajectories:

1. The default trajectory (dashed lines in Figure 2) which begins and develops naturally from Worsley's fix of 18 November.

2. A "nudged" trajectory leading from Worsley's 18 November fix to his 22 November fix. To achieve this, the simulated trajectory was co-located in the start point on 18 November and the averaged position offset for each time step added in such a way that the simulated position on 22 November matches Worsley's observation (dark orange and dark blue solid lines in Figure 2) This corresponds to a purely time dependent accumulating error.

3. A further alternative trajectory, nudged to align with Worsley's fix on 22 November only, without changing its general shape. This accounts for the possibility that the fix of 22 November is more accurate than the18 November fix (light orange and light blue solid lines in Figure 2).

Assessing the extremities described by each set of three trajectories allows us to estimate roughly the magnitude of position uncertainty associated with sea ice drift (see orange and blue ellipses in Figure 2).

### 3 Results and discussion

### 3.1 Estimating ERA-20C drift prediction error

To assess the relative uncertainty of the ERA-20C drift predictions in a more general sense (than only for the target period), we performed a basic assessment of mean predicted position error. Positions predicted by applying ERA-20C near-surface winds to virtual ice floes were reconstructed for the period 18 January 1915 until 21 November 1915, during which *Endurance*

was beset and drifting in the ice pack. The error is an average for the periods between positional fixes made by Worsley. The drift of virtual ice floes (defined by the navigational fixes) is simulated according to the method described in Section 2.4.2, using by ERA-20C winds, and wind factors and turning angles as used in simulation Cases 1-3. After sensitivity testing, these were decided to be:

Case 1: wind factor 1.75 %, turning angle 0°

Case 2: wind factor 1.85, turning angle 17.5°

Case 3: wind factor 2.5%, turning angle -25°

where a negative turning angle implies a deviation to the left of the wind. Whenever a position update from Worsley's log becomes available, the end position is automatically corrected, such that the initial position for the next drift step is Worsley's most recent fix. Mean error is computed as the mean of the distances between the end position from the forecast and the corresponding end positions available in Worsley's log. Figure 3 shows the histogram of all position errors for the period during which Endurance was beset and drifting in the pack ice. For Cases 1, 2 and 3 (representing different wind factor/turning angle combinations), mean position differences (i.e., the distance between simulated positions and Worsley's fixes) were 13.4, 14.0 and 15.4 km respectively. Median position differences were 10.7, 11.0 and 12.0 km respectively). Case 1 produces the lowest mean and median differences, though the cases produce generally similar error distributions.

### 3.2 Comparison of ERA-20C winds with Hussey observations

Figure 4 shows a comparison between Hussey's wind recordings and the ERA-20C wind data. Whilst there are broad similarities between the two datasets, there are differences in speed, direction, and timing which account for material differences in corresponding trajectories. Broadly, both datasets suggest strong north-component winds at the start of the target period, which weaken and veer to become light south-component winds and increase in strength slightly by the end of the period. Concerning changes in direction, however, Hussey observed an earlier and more gradual veering from northerly winds (to southerlies by the start of 20 November) than ERA, which suggests winds veered later and more suddenly to become south-south-easterly by mid-morning on 21 November. Thereafter, Hussey's recordings indicate winds remained roughly south-south westerly until the end of the period, with southerly and south-south-easterly variations for short periods. ERA winds remained more uniformly south-easterly until the end of the period. Concerning speeds, whilst both datasets agree on generally high speeds, followed by a decrease and then an increase, there are two principal discrepancies. The first is a significant difference between the mornings of 19 November and 20 November (up to 20 knots) due to Hussey's observation of a much faster speed drop following the strong northerlies (ERA winds stay stronger for longer and never drop quite as low as Hussey's recordings). The second is a significant discrepancy from the afternoon of 21 November until the end of the period. Whilst both datasets suggest winds of around 10 knots by the afternoon of 21, Hussey's observed gradual increase to the end of the period is preceded by an initial drop to below 5 knots. ERA does not produce this decrease, so whilst it shows a similar gradual increase through the end of the period, an discrepancy of 5-10 knots persists.

## 3.4 Reconstructed trajectories and sinking sites

For all three cases (which vary by wind factor and turning angle) using ERA-20C winds, the default trajectories (i.e., those starting at the 18 November position, indicated by dashed lines in Figure 2) yield the shortest distance between simulated and actual sinking sites (i.e., nudging the trajectories as explained in Section 2.5 did not improve simulated sinking locations).

Distances between the simulated and actual sinking locations for Cases 1-3 along these trajectories are 3.5, 0.0 and 10.8 km respectively. These simulated sinking locations are consistently north (by 1.7, 0.0 and 1.8 km) and east (by 3.0, 0.00 and 10.6 km) of the actual site.

Using Hussey's winds, Case 1 and 2 sinking locations are closest to the actual one when their trajectories are nudged to match both the 18 and 22 November positions. For Case 3, nudging to the 22 November produces the best result. Distances between

the simulated and actual sinking locations for Cases 1-3 along the above-mentioned trajectories are 0.3, 10.1 and 7.0 km respectively. Case 1's simulated sinking location is north (by 7.4 km) and east (by 5.6 km) of the actual location, whilst Cases 2 and 3's simulated sinking locations are north (by 7.6 and 5.9 km) and west (by 6.7 and 3.7 km) of the actual location.

All simulations, regardless of wind input data or parameter values, produce sinking locations with southerly component offsets from Worsley's estimate (consistent with the actual sinking location) and northerly component offsets from the actual location

(suggesting that with the exception of the idealised case, they do not quite capture the extent of the southerly excursion). These results, among others, are summarised in Table 1.

| | Case | Trajectory | Distance from Actual Sinking Location (km) | | | Distance from Worsley's Estimated Sinking Location (km) | | |
|---|---|---|---|---|---|---|---|---|
| | | | Total | Meridional | Zonal | Total | Meridional | Zonal |
| ERA-20C | 1 | Default (18 Nov) | 3.5 | 1.7 | 3.0 | 10.3 | -7.1 | 7.5 |
| | 2 | Default (18 Nov) | 0.0 | 0.0 | 0.0 | 9.9 | -8.8 | 4.5 |
| | 3 | Default (18 Nov) | 10.8 | 1.8 | 10.6 | 16.7 | -7.0 | 15.2 |
| Hussey | 1 | Nudged (18 & 22 Nov) | 9.3 | 7.4 | 5.6 | 1.9 | -1.5 | -1.1 |
| | 2 | Nudged (18 & 22 Nov) | 10.1 | 7.6 | -6.7 | 2.6 | -1.3 | -2.2 |
| | 3 | Nudged (22 Nov) | 7.0 | 6.0 | -3.7 | 5.7 | -5.6 | -0.6 |

**Table 1. Total and vector component distances from the various simulated sinking locations to the actual sinking locations, as well as to Worsley's estimated sinking location. Negative meridional (zonal) values indicate offsets to the south (west) of the reference.**

Within realistic parameter value ranges, applying ERA-20C to the drift simulation affords yields closer estimates of the sinking location. Though we are unable to say for sure, we deem Case 1 (Figure 2) to be the most likely, given that its parameter values are well within realistic ranges and result in the lowest mean and median error for the overall drift trajectory (January-November; Figure 3). In this case, a simulated sinking location some 3.5 km (1.7 km south, 3.0 km east) of the actual location is produced. In Case 2 (idealised case, Figure A3), where the simulated sinking location is forced to coincide with the actual

location, parameter values are still within realistic ranges as reported in the literature (1.85% and 17.5°), but further from their typical average values. To achieve the same using Hussey's observations, unrealistic parameter values are required (wind factors <3.8% and turning angles <-48°), which at the same time cause the corresponding ERA-20C simulations to be completely degraded (whereas for values optimised for ERA-20C, the Hussy results remain within the search area). This suggests ERA-20C wind inputs and resulting trajectories are more reliable.

In terms of the shape of the trajectory, all ERA-20C trajectories agree on a south-easterly excursion, followed by a clockwise turn to the north-west, prior to sinking. If Case 1 is the most likely and Case 2 is the idealised case, we deem Case 3 (Figure A4) a possible but relatively unlikely scenario. Acknowledging how widely parameter values vary, Case 3 is presented since it uses very typical, average values from the literature (wind factor 2.5 %, turning angle -25°). However, it does not produce very realistic sinking locations. It also produces higher mean and median error than Case 1 and 2.

For ERA-20C Case 1, the principal axis of uncertainty runs north-north-east to south-south-east (~ 140°). This is the same for Case 2 (idealised case; ~ 155°), and ESE for Case 3 (~ 122°). It is interesting to note that for many of the simulations, meridional and zonal offsets of sinking locations (representative of uncertainty in sea ice drift) are of the same order of magnitude as those associated with Worsley's traditional navigation methods, as analysed in detail by (Bergman and Stuart, 2018b). In some cases, they are nearly double.

**3.5 Accounting for discrepancies**

The accuracy of trajectories, as simulated in this study via a simplified free-drift method, depend on three main factors: the start points, the quality (resolution and accuracy) of the wind data and the selection of free-drift parameter values (though the latter two are probably more consequential). Since none of these are known with absolute certainty, the problem of reconstructing *Endurance's* trajectory is fundamentally under-constrained. Imposing assumptions allows us to close the

problem and draw inferences about the likely state of the other factors.

If we assume the wind-input data are perfect, remaining discrepancies between the simulated and actual sinking locations (and, by extension, errors in the shape of the associated trajectory) are likely due to the inaccuracy of the parameter values we impose (using, for example, values from the literature), which themselves depend on a host of factors. As a basic example, the more compact and thicker the sea ice, the larger the turning angle (Uotila et al., 2000; Martinson, Douglas G. Wamser, 1990), and

the rougher the floe, the greater the wind factor (Kottmeier et al., 1992). This is information we do not have.

Alternatively, if we force the parameter values to be "correct" (that is, tune the simulation to produce the correct sinking location as in ERA-20C Case 2), we may end up with values near their probable limits (or at least, more unusual according to

the literature). In this case, discrepancies between the imposed values and those we might have expected based on literature could be due to their needing to include, implicitly, effects not explicitly accounted for (e.g., internal ice stress and ocean
currents), or to inaccuracies of the wind data. For example, in Case 2, the perfect sinking location is produced by a wind factor of 1.85% and a turning angle of 17.5°. Whilst these are within empirical ranges, turning angles in the Weddell Sea are more usually negative (i.e., to the left of the wind). It is possible that the turning angle of 17.5° is required to mask an anticlockwise directional bias in the wind dataset of ~ 37.5°. In that case, the true turning angle becomes -20°, which would be very typical. Rapid changes in near-surface winds are often poorly reproduced by models, and since Endurance sank after the passage of a
cyclone, it's possible that this is the case. Moreover, such rapidly changing and gusty winds can cause the unpredictable breakup of sea ice, which might explain both the shift in ice conditions which catalysed the sinking of the vessel and the breakdown of the free-drift assumption (e.g. Nicolaus et al., 2022).

Whilst this sort of experimentation certainly yields insight, the selection of constraints and assumptions ultimately remains, to a certain extent, subjective.

**4 Conclusions**

This study demonstrates the potential of modern reanalysis weather models to help reconstruct possible ice drift trajectories of Shackleton's *Endurance*, and for use in marine archaeological projects more generally.

Whilst the prescription of a definitive trajectory is precluded by the sensitivity of simulations to choices of parameter values and potential inaccuracies of the wind data, a likely scenario was uncovered based on an envelope of results and consistent
features therein.

Specifically, we showed that between 18 and 22 November, *Endurance* likely followed a south-easterly excursion, followed by an anticlockwise turn and a short period of north-westward drift, prior to sinking, which is not described in Worsley's navigational data. The southerly excursion may have taken Endurance further south than the latitude at which the vessel was ultimately found.

We conclude that rigorous analysis of available weather and sea ice drift data is important to marine archaeological projects in sea ice covered oceans. This is not only true for proper positioning of the drifting survey vessel in the ice, but also for understanding the implications of sea ice drift on the position and trajectory of historic vessels locked in the ice. In this particular case, uncertainty due to the drift of sea ice was at least as large, and in many cases, larger than the uncertainty associated with navigational fixes.

**Appendix A1**

<Figures A1-A4>

## Author Contribution

Conceptualization: MdV; Data curation: MdV, CK, PK, JS; Formal analysis: MdV, PK, CK; Investigation: MdV, PK, CK, LR, JS; Methodology: MdV, CK, LR, PK; Project administration: MdV, CK, LR, JS; Resources: MdV, LR, PK; Software: MdV, CK, PK, MS; Supervision: MdV, JS, LR; Validation: CK, MdV; Visualization: MdV, CK, MS; Writing – original draft preparation: MdV, CK; Writing – review & editing: all authors.

## Competing interests

The authors declare that they have no conflict of interest.

## Data Availability

ERA-20C data is freely available at https://www.ecmwf.int/en/forecasts/datasets/reanalysis-datasets/era-20c. Leonard Hussey's meteorological observations are available upon request of the Archives of the Scott Polar Research Institute, University of Cambridge, with reference: SPRI Archive MS 1605/2/1 Hussey, L.D.A. Meteorological returns: Endurance 1.1.1915-31.12.1915.

## Acknowledgements

The authors would like to thank the following: The Falklands Maritime Heritage Trust for conceptualising and enabling this expedition and our participation in it; Director of Exploration for Endurance22, Mensun Bound, for encouraging the authors to publish our research; Naomi Boneham at the Archives of the Scott Polar Research Institute, University of Cambridge for the short-notice provision of Hussey's original meteorological records while we were at sea; Chairman of the Falklands Maritime Heritage Trust Donald Lamont and Director of Endurance22 Subsea Operations, Nico Vincent, for their valuable input to our manuscript.

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

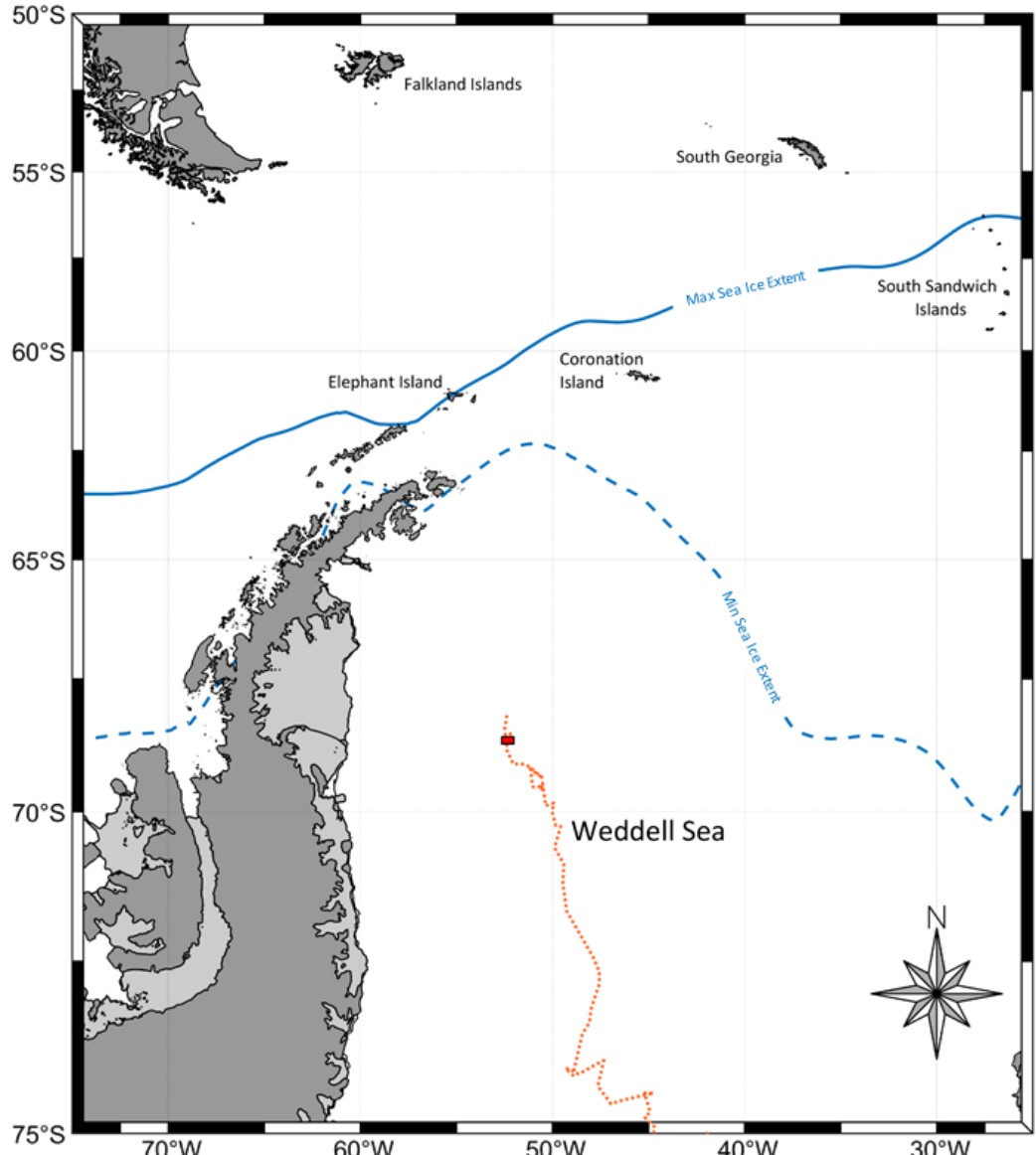

**Figure 1. Map showing the geographical context and selected key features of the study domain. The drift track of _Endurance_ prior to sinking is shown in orange, with the Endurance22 search area shown in red. Solid (dashed) blue lines indicate the long-term average maximum (minimum) sea ice extent. They are based on long-term averages (daily climatology) computed from the EUMETSAT OSI-SAF's Sea Ice Concentration Climate Data Record v3, accessed via the EU Copernicus CMEMS service (product code: OSISAF-GLO-SEAICE_CONC_TIMESERIES-SH-LA-OBS).**


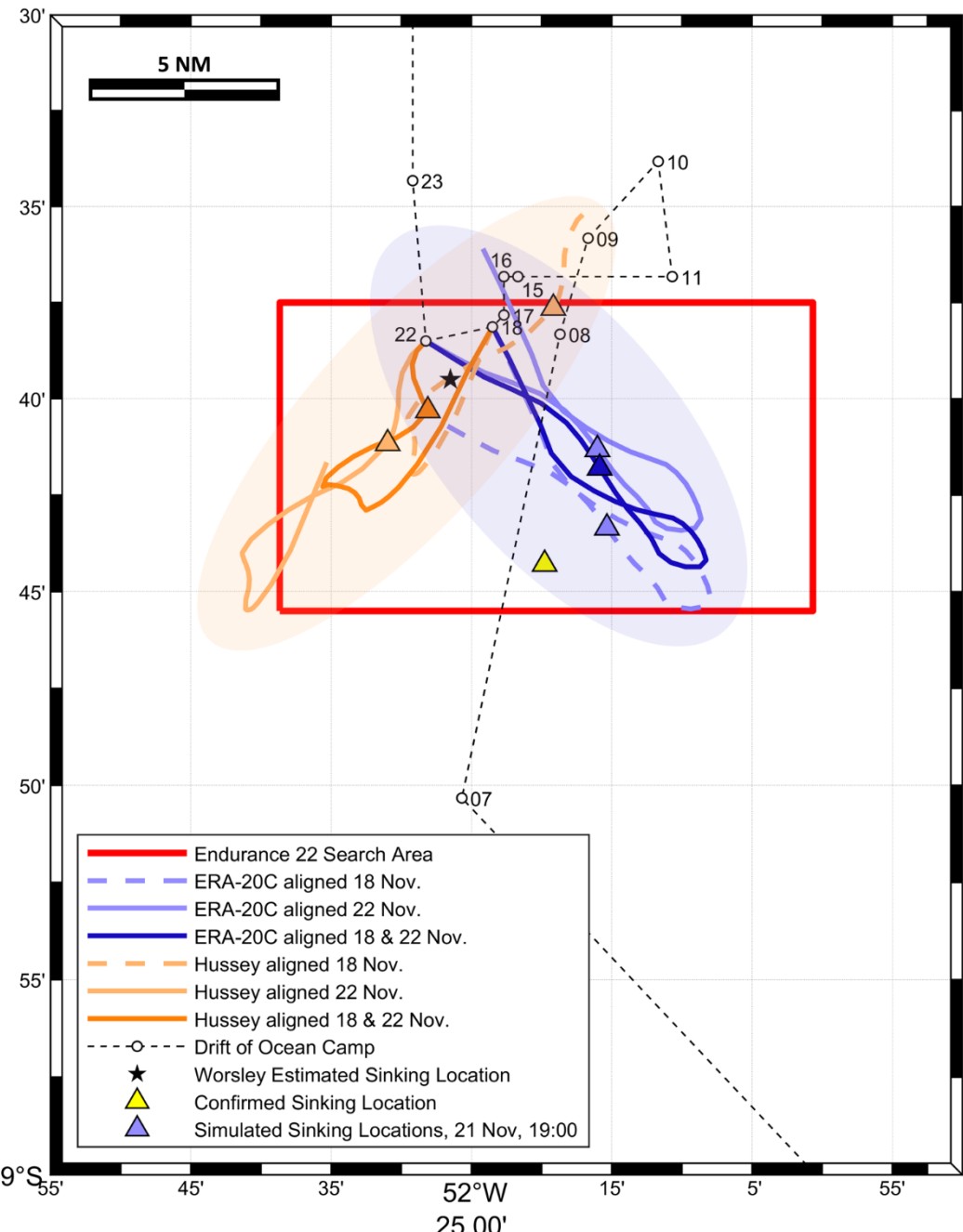

**Figure 2. Case 1 reconstructed drift tracks and sinking sites using ERA-20C reanalysis (blue) and Hussy's meteorological observations (orange). Case 1 utilised a wind factor of 1.75% and a turning angle of 0°. Coloured ellipses show approximate uncertainty regions associated with the respective dataset.**

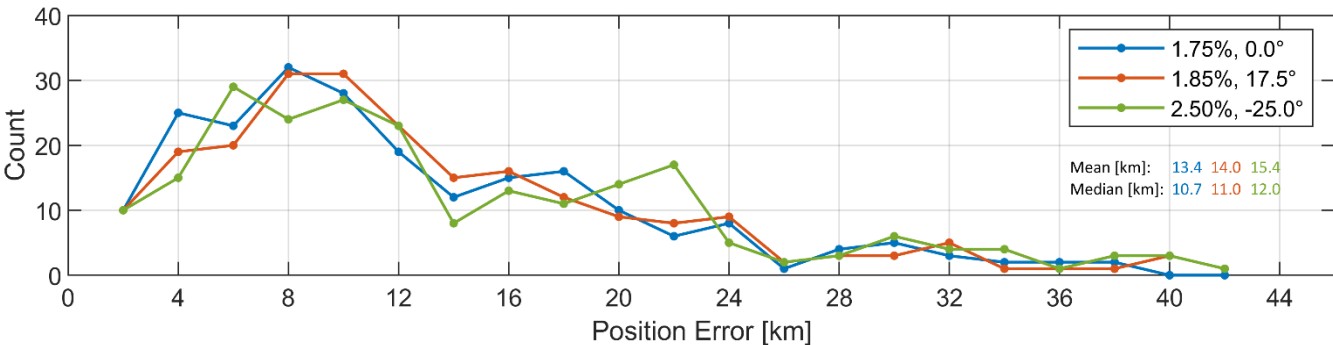

**Figure 3. Distribution of errors for predicted positions using initial positions from Worsley's navigational fixes and simulated ERA-20C surface wind-driven drift. The computation is for the period 18 January – 21 November 1915; the time during which *Endurance* was besets and drifting with the sea ice. The three sensitivity test cases discussed in Section 3.4 are presented (for each case, the legend refers to the wind factor and the turning angle). Also presented are mean and median errors.**

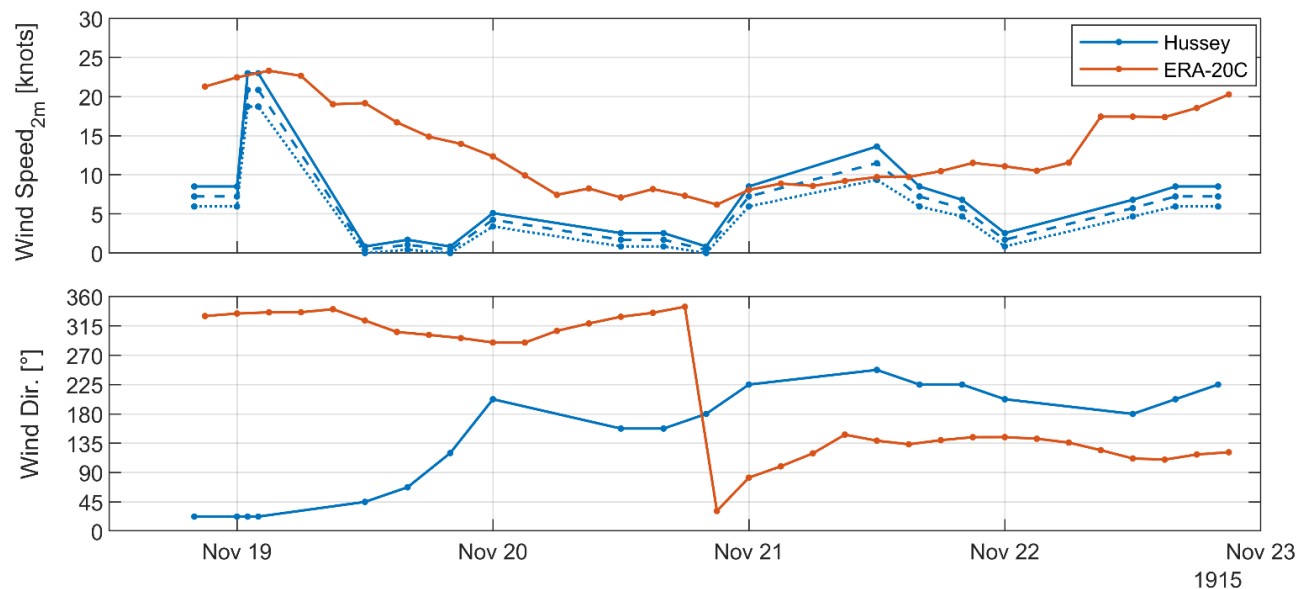

**Figure 4. Time series comparison of wind speeds (top panel) and directions (bottom panel) between recordings from Hussey and the ERA-20C product. For the Hussey wind speeds (since Hussey reported Beaufort indices), the solid (dotted) line indicates wind speeds corresponding to the upper (lower) Beaufort index bound. The dashed line shows the mean for that index.**

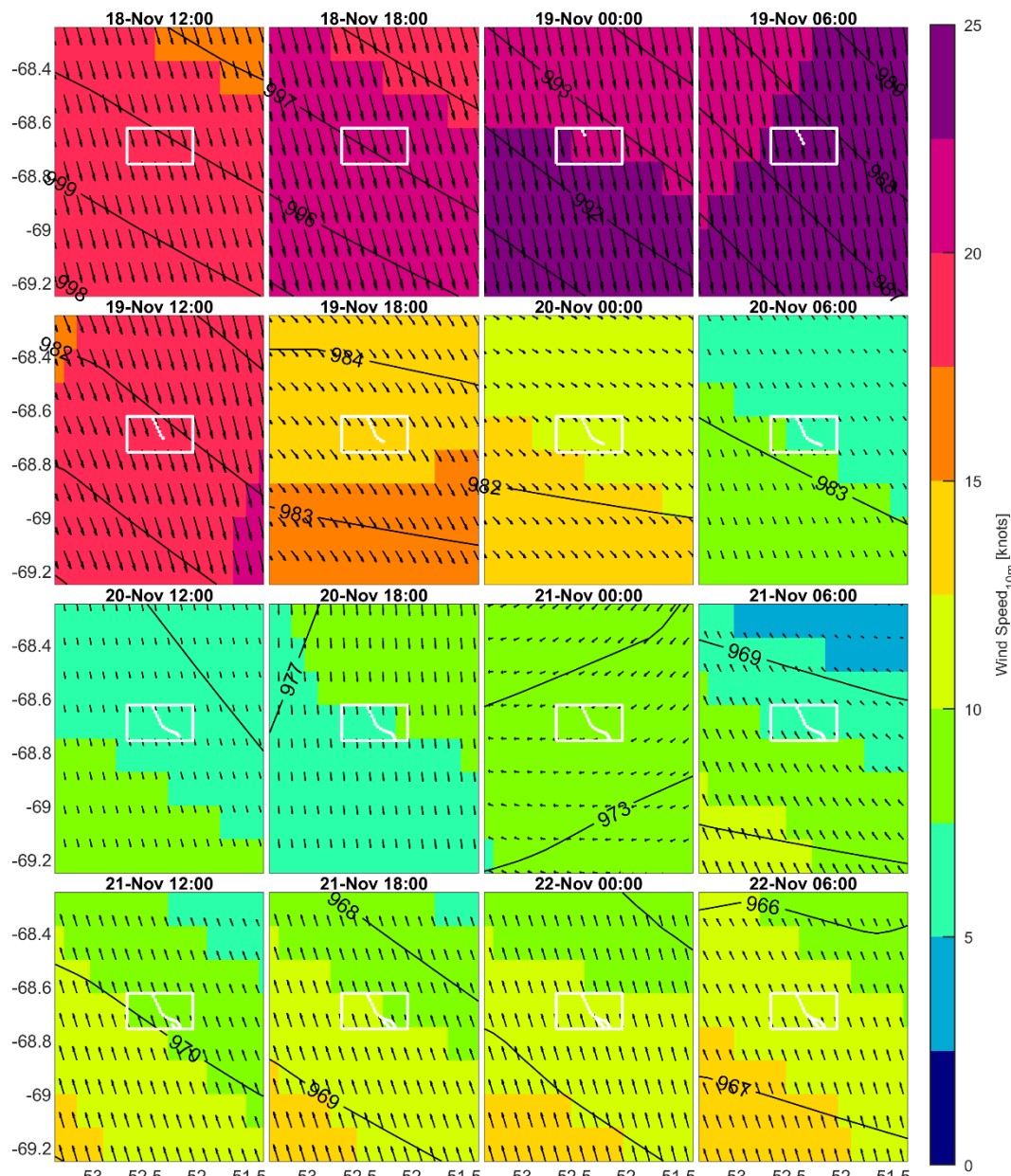

**Figure A1. 6-hourly maps of wind speed (colour scale, vector magnitude) and direction (vector orientation) and mean sea level pressure (contours) from ERA-20C. Also shown (white) are the search box and ERA-20C simulated trajectory. All dates refer to 1915.**

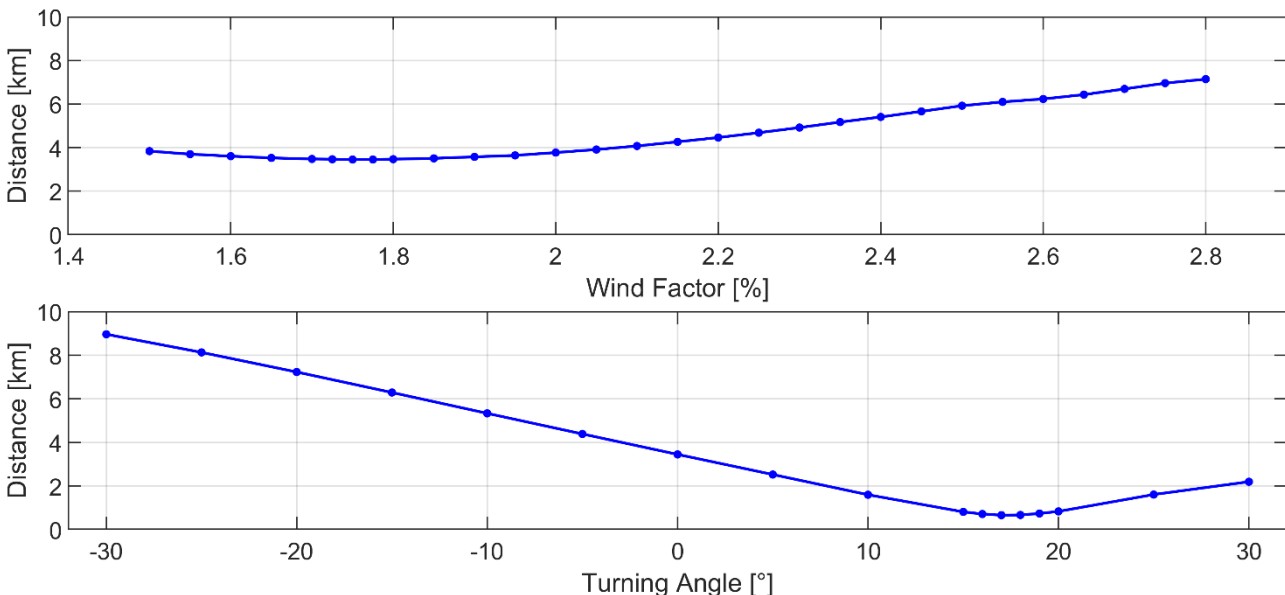

**Figure A2. Distance between ERA-20C simulated and actual sinking sites as a function of wind factor (top) and turning angle (bottom). These sensitivity results were used to arrive at the optimised parameter values for simulation Case 1 and the idealised values for Case 2.**


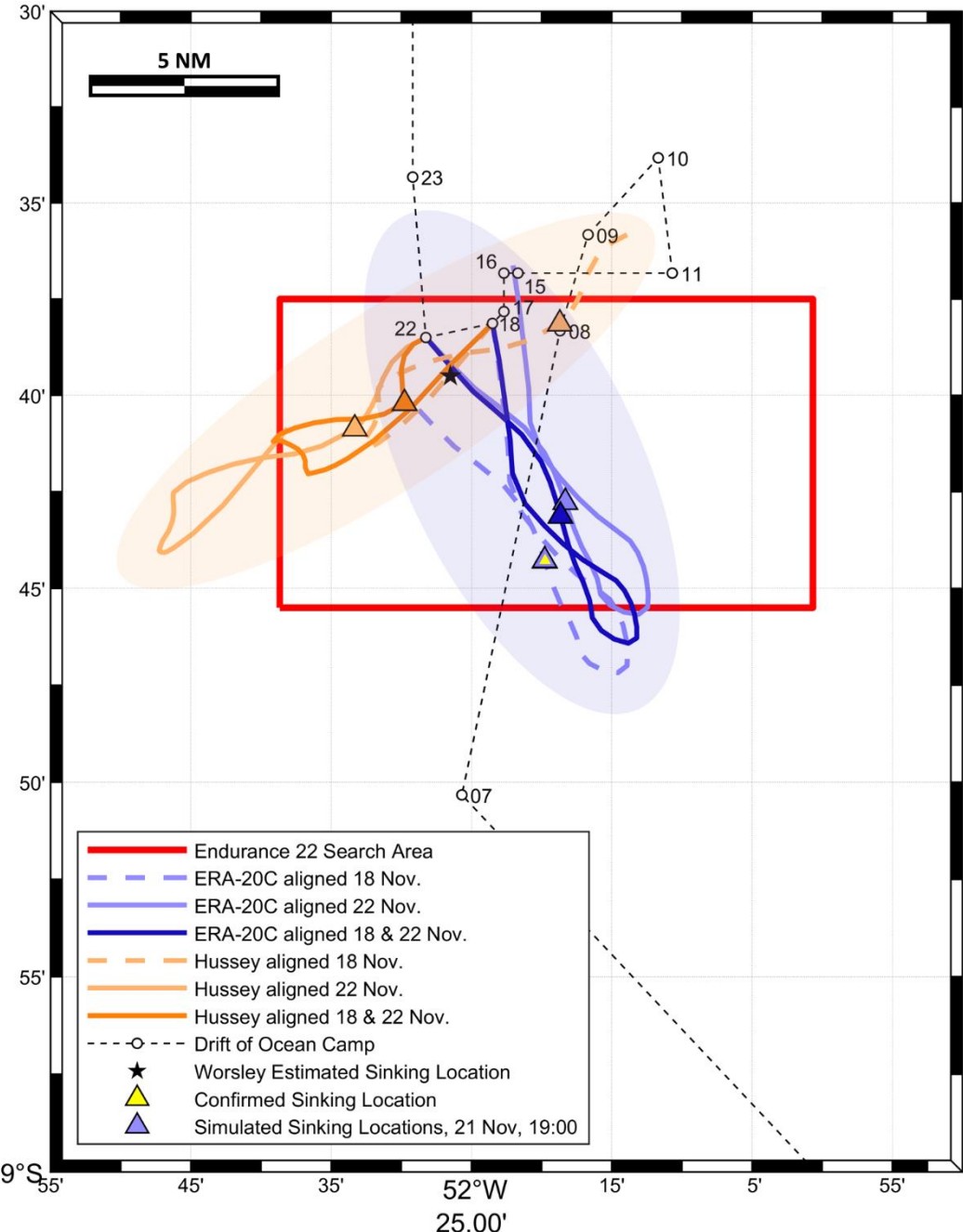

**Figure A3. Case 2 reconstructed drift tracks and sinking sites using ERA-20C reanalysis (blue) and Hussy's meteorological observations (orange). Case 2 is an idealised case, where the ERA-20C simulated sinking position is forced to coincide with the actual sinking location by adjusting model parameter values (note the coincident sinking location triangles). The required parameters are a wind factor of 1.85% and a turning angle of 17.5°. Coloured ellipses show approximate uncertainty regions associated with the respective dataset.**


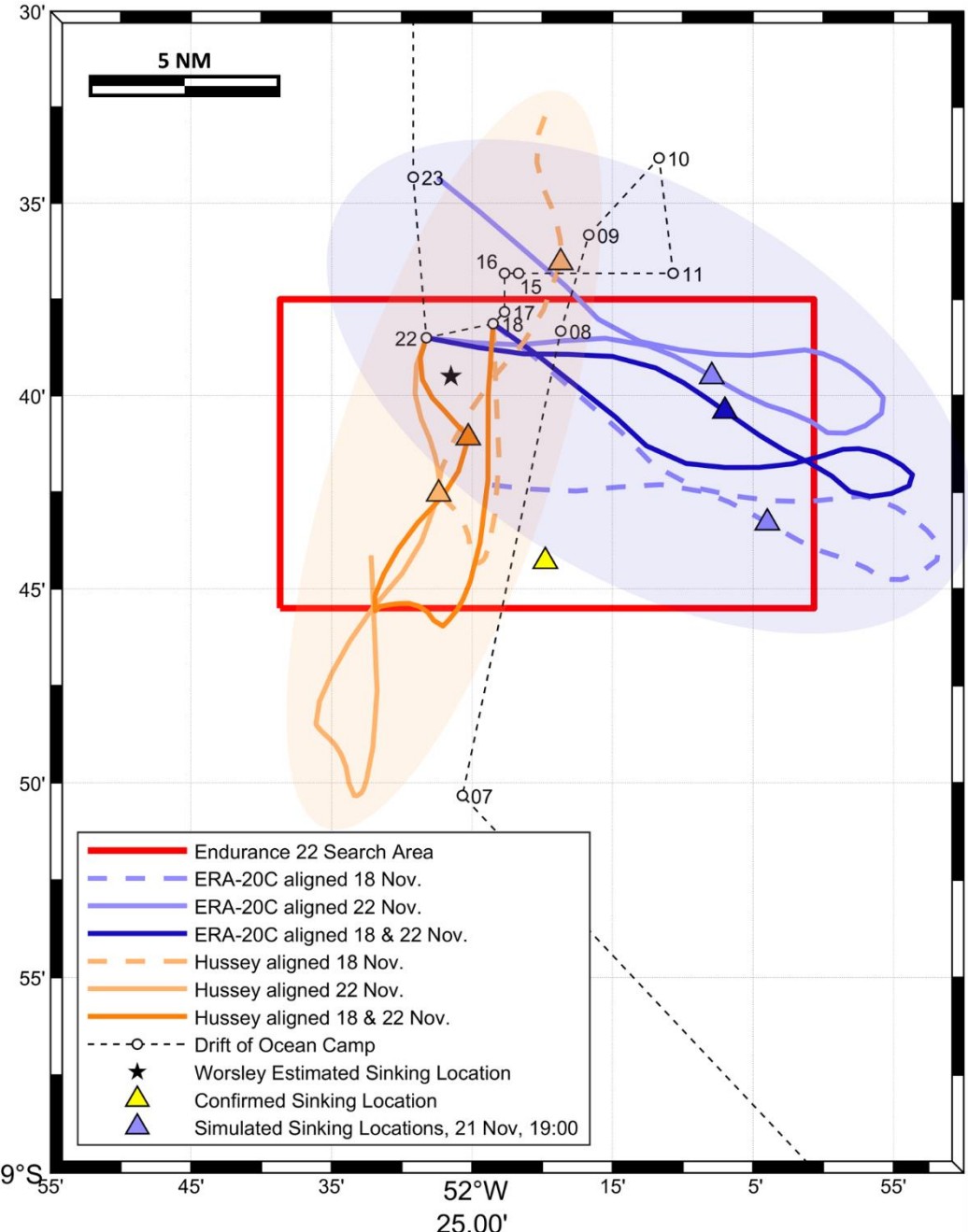

**Figure A4. Case 3 reconstructed drift tracks and sinking sites using ERA-20C reanalysis (blue) and Hussy's meteorological observations (orange). Case 3 utilised a wind drift factor of 2.5% and a turning angle of -25°. Coloured ellipses show approximate uncertainty regions associated with the respective dataset.**