# Peer review of "Understanding the drift of Shackleton's *Endurance* during its last days before it sank in November 1915 using meteorological reanalysis data"

_History of Geo- and Space Sciences, 2022_

## Referee Comment (RC1)

Review on the paper **hgss-2022-9** by de Vos et al. on the drift of Shackleton's Endurance

**General comment**.

The 20th century reanalyses by ECMWF (ERA-20C) and NOAA (20CR) provide detailed data on the structure of the atmosphere also for time periods, where global data are sparse, e.g at the end of the 19th century and the beginning of the 20th century. With these datastets it is now possible to analyse historical meteorological events, as has been done e.g. by Brönnimann et al. (2012, DOI:10.1127/0941-2948/2012/0337 ) for several storms since 1871. The present paper uses ERA-20C data for the reconstruction of trajectories of ice drift during the finale days of the famous Endurance expedition. Hence the paper fits quite nicely into the application of reanalysis data to historical events. Bevor publication (which is recommended), some modifications should be made according to the comments provided below.

**Specific comments**

Sec. 2.3:

1. According to Poli et al (2016), the horizontal resolution for ERA-20C is about 125 km. The area for the calculation of trajectories as shown in Fig. 1 is about 40x40 km. Hence the wind data are taken only from the grid point including the target area. Is this the case?

2. The wind observations by Hussey have been interpolated to hourly values (Sec. 2.2). Was the same interpolation also performed for the 3 hourly ERA data?

Sec. 2.4:

1. The drift trajectories are calculated from the wind data. But which method ist used for the calculation, e.g. a simple Euler forward time step or a more advanced integration/interpolation method?

2. Which time step is used for integration?

Sec. 3.2: Positions of the Endurance were predicted for the entire period. Please provide the beginning and end of this period.

Sec. 3.3:

1. In this section the winds obtained by ERA-20C and Hussey are compared indirectly by comparing trajectories which are an integrated form of winds. But the reader has no idea about the original wind data. Hence it is suggested, that the authors provide figures showing time series of wind speed and wind direction from both datasets , which is much more instructive for the comparison.

2. When comparing wind data from different sources one has to make sure, that these are taken at the same height above ground or have been interpolated to the same height. At which heights are the winds from Hussey and ERA-20C taken?

A note:

Data from NOAA-20CR reanalysis have been used by Etling (2017, DOI: 10.1127/metz/2017/0853) for investigating the atmosperic conditions of another famous polar expedition, the balloon flight of Andre and his crew in order to reach the North Pole in 1897 (which failed).

---

## Author Comment (AC1)

Review on the paper **hgss-2022-9** by de Vos et al. on the drift of Shackleton's Endurance

General comment.

The 20th century reanalyses by ECMWF (ERA-20C) and NOAA (20CR) provide detailed data on the structure of the atmosphere also for time periods, where global data are sparse, e.g at the end of the 19th century and the beginning of the 20th century. With these datastets it is now possible to analyse historical meteorological events, as has been done e.g. by Brönnimann et al. (2012, DOI:10.1127/0941-2948/2012/0337 ) for several storms since 1871. The present paper uses ERA-20C data for the reconstruction of trajectories of ice drift during the finale days of the famous Endurance expedition. Hence the paper fits quite nicely into the application of reanalysis data to historical events. Bevor publication (which is recommended), some modifications should be made according to the comments provided below.

The authors would like to thank the reviewer for this positive and constructive input. Amendments made in response have both strengthened the manuscript directly and prompted us to enhance other aspects of the work. As the corresponding author I would also like to apologise for the delayed response. The timing of the review coincided with an Antarctic voyage, as well as other end-of-year academic commitments, thereby delaying the submission of these responses.

**Specific comments**

Sec. 2.3:

1.According to Poli et al (2016), the horizontal resolution for ERA-20C is about 125 km. The area for the calculation of trajectories as shown in Fig. 1 is about 40x40 km. Hence the wind data are taken only from the grid point including the target area. Is this the case?

Yes, this is the case. However, when downloading products, ECMWF makes information available at a range of regular grid resolutions via its Meteorological Interpolation and Regridding (MIR) package. Whilst no additional dynamical information is available simply by downloading data at higher interpolated resolution, the 0.125° (approx. 13.9 km) product was selected in case it facilitated somewhat smoother drift stepping. Figure 1 below shows the forcing data and search area, giving some idea of the resolution (one wind arrows for each grid cell, with cells also visible in the shading) Notwithstanding, comparing results using the 0.5° and 0.125° resolution revealed negligible differences.

In response to a call for the resolution of the data to be provided, the following text has been added to Section 2.3:

*"ERA-20C has a spatial resolution of approximately 125 km on its native triangular grid (Poli et al., 2016b). However, interpolated data were downloaded on a regular grid with a resolution of 0.125. This interpolated product is produced by ECMWF's Meteorological Interpolation and Regridding (MIR) package and is available via ECMWF's download portal at: https://apps.ecmwf.int/datasets/data/era20c-daily/levtype=sfc/type=an/."*

[Figure]

**1915-11-21 18:00**

*Figure 1. A snapshot of wind speed (shading & arrow length) and direction (arrow orientation) and mean sea level pressure (isobars) from ERA-20C. The search area is shown in red. There is one wind arrow per grid cell, giving an indication of the spatial resolution of the forcing product. Note that this is an interpolation onto a 0.125° regular grid from the native grid output from ERA-20. It is made available for download at this resolution via ECMWF's website.*

2.The wind observations by Hussey have been interpolated to hourly values (Sec. 2.2). Was the same interpolation also performed for the 3 hourly ERA data?

We thank the reviewer for picking this up. Both sets of trajectories (ERA and Hussey) were computed at both 1 and 3-hourly resolution, producing negligible differences. However, it's true that the temporal resolution of the manuscript results should be consistent. We have thus changed the Hussey trajectories to 3-hourly to match the ERA ones.

Sec. 2.4:

1.The drift trajectories are calculated from the wind data. But which method is used for the calculation, e.g. a simple Euler forward time step or a more advanced integration/ interpolation method?

Two methods were tested. At each time step, the new position is computed from the start position by supplying the drift direction and distance to:
1.  The Vincenty formula (Vincenty, 1975) via the *m_fdist.m* function for MATLAB by Pawlowicz (2020)
2.  A simple trigonometric approach via the *ptlatlon.m* function for MATLAB by (Marty, 2018). Specifically:

$$lat_{end} = \text{asin} \left(\sin(lat_{start}) \cos\left(\frac{drift\_dist}{R_{earth}}\right) + \cos(lat_{start}) \sin\left(\frac{drift\_dist}{R_{earth}}\right) \cos\left(drift\_bearing\right)\right)$$

$$lon_{end} = lon_{start} + atan2(\sin\left(drift_{bearing}\right) \sin\left(\frac{drift\_dist}{R_{earth}}\right) \cos(lat_{start}) \cos\left(\frac{drift\_dist}{R_{earth}}\right) - \sin(lat_{start}) \sin\left(lat_{end}\right)$$

Both methods produce nearly identical trajectories

2.Which time step is used for integration?

1-hourly and 3-hourly were tested. For the ERA, the 1-hourly represents an interpolated step, as the native resolution is 3-hourly. For the Hussey trajectories, both the 1 and 3-hourly are interpolated steps, as they are daily (mostly noon) observations. Since the trajectories were insensitive to the choice of time step, all trajectories presented in the manuscript utilised a 3-hourly time step.

Sec. 3.2: Positions of the Endurance were predicted for the entire period. Please provide the beginning and end of this period.

The period for which the drift simulation error was assessed was 18$^{th}$ January 1915 until 21$^{st}$ November, 1915. This is the period from when Endurance became beset until the time of the sinking. The text shall be amended to include this information.

Sec. 3.3:

1.In this section the winds obtained by ERA-20C and Hussey are compared indirectly by comparing trajectories which are an integrated form of winds. But the reader has no idea about the original wind data. Hence it is suggested, that the authors provide figures showing time series of wind speed and wind direction from both datasets , which is much more instructive for the comparison.

We thank the reviewer for this valuable suggestion. Timeseries graphs of wind speed and direction have been prepared and shall be included in the revised manuscript. This will also help to elucidate the differences between the ERA-20C and Hussey trajectories.

2.When comparing wind data from different sources one has to make sure, that these are taken at the same height above ground or have been interpolated to the same height. At which heights are the winds from Hussey and ERA-20C taken?

This is a valuable comment. In the initial set of results, the 10 m winds from ERA-20C were used without correction (as were the Hussey observations). We have now interpolated both the ERA-20C winds and Hussey's observed winds to the 2 m level by means of a logarithmic wind profile correction (Manwell et al., 2009), prior to use in the drift calculations. The choice of 2 m is a best guess to try and capture the winds which the floes of variable thickness (and irregular ridges) might have been forced by.

A note:
Data from NOAA-20CR reanalysis have been used by Etling (2017, DOI: 10.1127/metz/2017/0853) for investigating the atmosperic conditions of another famous polar expedition, the balloon flight of Andre and his crew in order to reach the North Pole in 1897 (which failed)

References

Manwell, J., McGowan, J., and Rogers, A.: Wind energy explained: theory, design, and application, 2nd ed., John Wiley & Sons Ltd., Chichester, 677 pp., 2009.

Pawlowicz, R.: M_Map: A mapping package for MATLAB, https://www.eoas.ubc.ca/~rich/map.html, 2020.

Marty, R.: ptlatlon, https://nl.mathworks.com/matlabcentral/fileexchange/67254-ptlatlon, 2018.

Vincenty, T.: Direct and Inverse Solutions of Geodesics on the Ellipsoid with Application of Nested Equations, 88–93 pp., 1975.

---

## Author Comment (AC3)

This paper presents an interesting new reanalysis of meteorological data that augments and complements previous studies on the drift of the Endurance prior to the vessel's sinking.

The following are relatively minor comments:

The authors would like to thank the commenter for the interest shown and the effort in conveying this constructive input. We endeavour to incorporate this input in the revised manuscript. As the corresponding author I would also like to apologise for the delayed response. The timing of the review coincided with an Antarctic voyage, as well as other end-of-year academic commitments, thereby delaying the submission of these responses.

Line 34: "low frequency" - for posterity, as I cannot find the detail elsewhere, please include the type and actual frequencies of the sidescan sonar. While its frequency may be considered "low frequency" on the radio frequency spectrum, for underwater acoustics it is likely to be either medium or high frequency, that is, above 20 kHz.

This is a very valid point as regards "low frequency". The SAAB Sabretooth AUVs were fitted with Edgetech 2105 side scan sonar systems, operating at frequencies of 75, 230 or 410 Khz (Gilbert, 2021). This detail will be added to the revised manuscript.

Line 52: "local time" - it is only near the end of the paper than the reader finds the longitude and so is able to interpret local time. The nuances between Zone Time (integer hour offset from GMT (UTC)) and Ship's Time, and the relationship of Ship's Time to Local Apparent Noon on the Endurance are discussed in Bergman and Stuart (2019). This paper also gives insights into the accuracy of navigational sights during earlier parts of the voyage.

In response to a review comment, a context map with clear longitudinal information has been prepared and will be included in the revised manuscript. This should assist. However, information about the time standards used will also be added to the methodology section (i.e., the use of ERA-20C data for the simulation, which is referenced to UTC).

Line 53: There is no attempt to quantify what is meant by accurate. Perhaps a reading of, and reference to, Bergman and Stuart may help. Also affecting accuracy may be the reanalysis using modern lunar ephemerides and catalogues of star positions in the unpublished paper by Bergman et al. available at http://fer3.com/arc/imgx/OccultationCEPreprint.pdf

Field Code Changed

We thank the commenter for this information. We will review the suggested sources and clarify or modify the notion of accuracy in the revised manuscript.

Line 136: Multiplying the 24-hour error range of 4 km to 10 km by four for the 4 day period is too simplistic. It would be a fair approximation if and only if there was no change in direction for the drift over the 4 days. The error per day should be treated as a vector and not a scalar and the 4-dat vector error estimated.

We thank the commenter for this very valid comment, with which we agree. We have removed the estimation of the accumulated error over the target period, since we have no way of computing it objectively given the lack of position recording during these 3-4 days. Previously it was simplistically computed based on the mean daily error which we calculated for the period 18 January – 21 November, from simulated positions and positions logged by Worsley. The

single vector errors in sinking position along the various simulated and nudged trajectories is reported, as well as the mean error for the period 18 January – 21 January based calculations made whenever possible with respect to position recordings.

Bergman, L. and Stuart, R.G., 2019. Navigation on Shackleton's voyage to Antarctica'. *Records of the Canterbury Museum*, *33*, pp.5-22

References

Gilbert, N. (2021). *Endurance 22 Initial Environmental Evaluation*. 21/12/2022https://endurance22.org/uploads/2022/01/Endurance22_IEE.pdf

---

## Author Comment (AC4)

The authors present work that describes how they have used reanalysis data to better understand the sea ice conditions and trajectory before the Endurance sank in the Weddell Sea in November 1915. They find that the trajectory using reanalysis data has a more accurate wreck site to the actual location as compared to trajectories using observational data from the Endurance crew. Overall, this study is novel in that it presents some new methods for Polar marine archeology and how to use new data. I have some moderate to minor concerns that I think should be addressed before this is accepted for publication, including addition of another figure.

*The authors would like to thank the reviewer for this positive and constructive input. Amendments made in response have both strengthened the manuscript directly and prompted us to enhance other aspects of the work. As the corresponding author I would also like to apologise for the delayed response. The timing of the review coincided with an Antarctic voyage, as well as other end-of-year academic commitments, thereby delaying the submission of these responses.*

Specific comments:

- Introduction: It would be nice to include details about the 1914-1915 expedition for those readers unfamiliar with it. For example, it's worth mentioning briefly that the crew all survived and was rescued from Elephant Island and how that compares in location to the wreck. Additionally, it would be useful to summarize other expeditions searching for the Endurance and how the successful approach in 2022 was different from those ventures. Finally, I didn't really understand how the sinking happened and it would be useful to clarify: was the vessel stuck in the ice and drifted several days before sinking or did it sink immediately? Could this account for some of the poor estimate in location from Worsley's location?

  *We thank the reviewer for this valuable insight and on reflection, it is certainly helpful to any reader who may not be well-versed to be presented at least some basic context.*

  *Further contextual details (and specifically those suggested) shall be included in the revision. Further, since the more detailed analysis conducted as part of this review process has elucidated the many possible sources of error, we will include not only some text regarding the sinking and possible reasons for Worsley's position estimate, but also sources of uncertainty generally as regards our simulation estimates.*

- Line 56 – Please clarify in text (possibly in the introduction) what "Ocean Camp" is and how that differs from the location of the Endurance.

  *The following text (or similar) is the be added to the Introduction, where the enhanced details about the Imperial Trans-Antarctic Expedition are to be included (see previous comment).*

  *"After Endurance became too badly damaged to be used as the team's primary shelter, the team abandoned the vessel, intending to begin a march to the west. After little more than a week, however, the march was abandoned after harsh conditions greatly hampered progress. Instead, the team would camp on an ice floe until sea ice conditions became more conducive to moving. The floe, located a short distance from the wreckage of the Endurance, was named Ocean Camp by the team."*

- Line 58 – Why did Worsley add further offset? Is this known? Is the star on Figure 1 showing Worsley's location of the sinking include the offset?

It is not known exactly why Worsley added this offset. It may be that Worsley computed applied some kind of integration of their position change based on his estimate of the sea ice drift, though this is not known with certainty (and also not backed up by Hussey's wind observations). Bergman & Stuart (2018) suggest that this was simply an offset added to the nearest dependable fix (22nd Nov.) available to Worsley.

- Line 78 – Please give the horizontal resolution of the ERA20 data (1degree? Higher resolution?) as this is relevant for how well it can resolve sea level pressure fields and near surface winds.

  The following text shall be added to Section 2.3:

  "*ERA-20C has a spatial resolution of approximately 125 km on its native triangular grid* (Poli et al., 2016). *However, interpolated data were downloaded on a regular grid with a resolution of 0.125. This interpolated product is produced by ECMWF's Meteorological Interpolation and Regridding (MIR) package and is available via ECMWF's download portal at: https://apps.ecmwf.int/datasets/data/era20c-daily/levtype=sfc/type=an/.*"

- Line 93 and conclusion – You should mention that free drift is reasonable over short time scales *in the Antarctic* only - Kwok et al. 2017 is a good source for this (which is already in your reference list). This is relevant because you can't make the same assumptions in the Arctic (e.g. to have found the Erebus and Terror from the Franklin Northwest Passage Expedition). This is relevant because one of your main conclusions is that marine archaelogy in sea ice covered oceans can benefit from drift data, but the hemisphere may affect this technique. If you know of other Antarctic vessels that might benefit from this technique it would be useful to list them in the introduction or conclusion.

  We thank the reviewer for this comment as the assumption of free-drift, and the tuning of free-drift parameters is certainly a key issue which materially affects the outcome of this study (and not only the general application of this method to marine archaeology). It prompted us to enhance our approach as explained below.

  A paragraph explaining in detail the strengths, weaknesses, assumptions and implications of the free-drift approach, and a review of free-drift parameter settings is to be included in Section 2.4. The text draws on the findings of (Doble & Wadhams, 2006; Kottmeier et al., 1992; Nakayama et al., 2012; Nie et al., 2022; Uotila, 2001; Uotila et al., 2000; Vihma et al., 1996; Vihma & Launiainen, 1993; Womack et al., 2022).

  A further reason for including this analysis is because after further review of the above literature, it became clear that whilst the free-drift assumption is reasonable for the domain in question, and that wind is the primary forcing, drift trajectories are particularly sensitive to free-drift parameter settings (wind drag and turning angle). In particular, it is apparent that parameter settings derived from buoy experiments in the literature vary widely in space, time and even from instrument to instrument in roughly the same area. As Kottmeier et al. (1992) put it: *"Parameters change both from one data set (buoy, period, region) to the other and within a certain data set."* This prompted us to change our approach from simply reusing the free-drift configuration settings used onboard during the Endurance22 expedition, to a more considered, comprehensive approach whereby setting were tuned in a series of sensitivity analyses. An example of a sensitivity test is shown in Figure 1 below. Given the expanded envelope of results, in Section 2.4 and 2.5, we will therefore describe the range of possible outcomes using different settings, thereby outlining likely scenarios (i.e., consistent aspects of the unknown drift) as well as remaining uncertainty.

[Figure]

*Figure 1. An example of a sensitivity test which shows how the distance of the ERA and Hussey simulated sinking position varies as a function of wind drag parameter selection. For each of the ERA and Hussey simulations, results for the trajectories aligned at the 18th, 22nd and both the 18th and 22nd are shown (e.g., HUSS18&22 is the distance of the sinking site from the actual wreck site according to the Hussey-simulated trajectory aligned at both the 18th and 22nd).*

- Line 144-146: It's interesting that ERA20 aligned to Nov.22, 1915 produced the most accurate wreck location. In addition to your possible explanations about only having 12 hrs of observations in a day from the logs and possible not free drift, I think you should mention that it's possible that if the sinking happened during/after a storm (cyclone) then changes in near-surface wind gustiness and direction, which are notoriously poor in models and short lived, could have been relevant and caused the pack to break up in sometimes not predictable ways. This has happened for recent voyages (see. Nicolaus et al. 2022, "Overview of the MOSAiC Expedition: Snow and sea ice", doi: https://doi.org/10.1525/elementa.2021.000046) where sea ice deformation and motion during a storm was certainly not in free drift.

  First, we qualify this finding in light of the additional work which we have done and changes we have made to the simulation configuration. Further, we thank the reviewer for this excellent suggestion and shall include text to explain this in Section 3.3. This also ties into an expanded explanation which we are to include about the discrepancies between the Hussey and ERA-20C wind directions. We explain this in the context of how we arrived at out optimal free-drift parameter settings. For example, it may be that the ERA-20C winds have an anti-clockwise bias, since Hussey's winds are quite significantly rotated clockwise relative to ERA-20C (though are of course also subject to error) and the optimal turning angle is smaller than what we might expect from (some of) the literature.

- Comments on Figures:
- A map of the region would be very helpful showing at minimum the Weddell Sea, Elephant Island, and S. Georgia Island. This could be combined with Figure 1 perhaps.

  We thank the reviewer for the suggestion and agree, this would aid in readability. A context map shall be added and first referred to in Section 1. A preliminary map has been prepared (Figure 2).

[Figure]

*Figure 2. A map showing the context of the study, to be included in the revised manuscript.*

- Figure 1 – it appears the point 10 is off the map. Having a compass for directions would be helpful for immediate orientation. It's worth mentioning as well that the actual wreck is well within the uncertainty region for the ERA winds but right on the edge of the uncertainty region with observed winds.

  This figure's map boundaries have been revised so that the position for the 10th is not cut off, and a compass pointer added. Commentary will be added to the manuscript which makes reference to the wreck location in relation to the trajectory envelopes.

- I think it would be useful to have ERA20 sea level pressure maps at 12GMT from Nov.18-22 showing both pressure contours and wind vectors in relation to where the ship was, roughly. It would also be useful to list the modeled wind speed and direction at the ship's location and the observations at that time. Those values could be listed in the panels for each day. This will help the readers understand how different the model is from observed winds and how the local, observed wind field may have differed from the large scale flow.

  Maps of mean sea level pressure (contours), wind speed (shading, vector length) and direction (vector orientation) which also show the search area and Endurance trajectory have been created. Figure 3 below is an example. Select snapshots which summarise conditions during the period shall be selected for inclusion in the revised manuscript.

  In response to another review comment, time series graphs of ERA-20C and Hussey's observed wind data have been created to facilitate easy comparison. We trust that this satisfies the second part of this comment.

Preliminary examples of the wind maps (Figure 3) and timeseries graphs (Figure 4) to be included in the revised manuscript are shown below.

[Figure]

*Figure 3. A snapshot of wind speed (shading & arrow length) and direction (arrow orientation) and mean sea level pressure (isobars) from ERA-20C. The search area and target period drift trajectory at that point in time are shown in white.*

[Figure]

*Figure 4. A time series comparison of wind speeds and directions observed by Hussey and those produced by the ERA-20C reanalysis.*

References

Bergman, L., & Stuart, R. G. (2018). Navigation of the Shackleton Expedition on the Weddell Sea pack ice. *Records of the Canterbury Museum*, *32*, 67–98.

Doble, M. J., & Wadhams, P. (2006). Dynamical contrasts between pancake and pack ice, investigated with a drifting buoy array. *Journal of Geophysical Research: Oceans*, *111*(11). https://doi.org/10.1029/2005JC003320

Kottmeier, C., Olf, J., Frieden, W., & Roth, R. (1992). Wind forcing and ice motion in the Weddell Sea region. *Journal of Geophysical Research*, *97*(D18). https://doi.org/10.1029/92jd02171

Nakayama, Y., Ohshima, K. I., & Fukamachi, Y. (2012). Enhancement of sea ice drift due to the dynamical interaction between sea ice and a coastal ocean. *Journal of Physical Oceanography*, *42*(1), 179–192. https://doi.org/10.1175/JPO-D-11-018.1

Nie, Y., Uotila, P., Cheng, B., Massonnet, F., Kimura, N., Cipollone, A., & Lv, X. (2022). Southern Ocean sea ice concentration budgets of five ocean-sea ice reanalyses. *Climate Dynamics*, *59*(11–12), 3265–3285. https://doi.org/10.1007/s00382-022-06260-x

Poli, P., Hersbach, H., Dee, D. P., Berrisford, P., Simmons, A. J., Vitart, F., Laloyaux, P., Tan, D. G. H., Peubey, C., Thépaut, J. N., Trémolet, Y., Hólm, E. v., Bonavita, M., Isaksen, L., & Fisher, M. (2016). ERA-20C: An atmospheric reanalysis of the twentieth century. *Journal of Climate*, *29*(11), 4083–4097. https://doi.org/10.1175/JCLI-D-15-0556.1

Uotila, J. (2001). *Observed and modelled sea-ice drift response to wind forcing in the northern Baltic Sea*. *53*, 112–128.

Uotila, J., Vihma, T., & Launiainen, J. (2000). Response of the Weddell Sea pack ice to wind forcing. *Journal of Geophysical Research: Oceans*, *105*(C1), 1135–1151. https://doi.org/10.1029/1999jc900265

Vihma, T., & Launiainen, J. (1993). Ice drift in the Weddell Sea in 1990-1991 as tracked by a satellite buoy. *Journal of Geophysical Research*, *98*(C8). https://doi.org/10.1029/93jc00649

Vihma, T., Launiainen, J., & Uotila, J. (1996). Weddell Sea ice drift: Kinematics and wind forcing. *Journal of Geophysical Research: Oceans*, *101*(C8), 18279–18296. https://doi.org/10.1029/96JC01441

Womack, A., Vichi, M., Alberello, A., & Toffoli, A. (2022). Atmospheric drivers of a winter-to-spring Lagrangian sea-ice drift in the Eastern Antarctic marginal ice zone. *Journal of Glaciology*, *68*(271), 999–1013. https://doi.org/10.1017/jog.2022.14

---

## Author Response (AR1)

**Response to RC1**

Review on the paper hgss-2022-9 by de Vos et al. on the drift of Shackleton's Endurence

**General comment.**

The 20th century reanalyses by ECMWF (ERA-20C) and NOAA (20CR) provide detailed data on the structure of the atmosphere also for time periods, where global data are sparse, e.g at the end of the 19th century and the beginning of the 20th century. With these datastets it is now possible to analyse historical meteorological events, as has been done e.g. by Brönnimann et al. (2012, DOI:10.1127/0941-2948/2012/0337) for several storms since 1871. The present paper uses ERA-20C data for the reconstruction of trajectories of ice drift during the finale days of the famous Endurance expedition. Hence the paper fits quite nicely into the application of reanalysis data to historical events. Bevor publication (which is recommended), some modifications should be made according to the comments provided below.

The authors would like to thank the reviewer for this positive and constructive input. Amendments made in response have both strengthened the manuscript directly and prompted us to enhance other aspects of the work. As the corresponding author I would also like to apologise for the delayed response. The timing of the review coincided with an Antarctic voyage, as well as other end-of-year academic commitments, thereby delaying the submission of these responses.

**Specific comments**

**Sec. 2.3:**

1.According to Poli et al (2016), the horizontal resolution for ERA-20C is about 125 km. The area for the calculation of trajectories as shown in Fig. 1 is about 40x40 km. Hence the wind data are taken only from the grid point including the target area. Is this the case?

Yes, this is the case. However, when downloading products, ECMWF makes information available at a range of regular grid resolutions via its Meteorological Interpolation and Regridding (MIR) package. Whilst no additional dynamical information is available simply by downloading data at higher interpolated resolution, the 0.125° (approx. 13.9 km) product was selected in case it facilitated somewhat smoother drift stepping. Figure 1 below shows the forcing data and search area, giving some idea of the resolution (one wind arrows for each grid cell, with cells also visible in the shading) Notwithstanding, comparing results using the 0.5° and 0.125° resolution revealed negligible differences.

In response to a call for the resolution of the data to be provided, the following text has been added to Section 2.3:

"Spatial resolution is approximately 125 km on the native ERA-20C triangular grid (Poli et al., 2016). However, interpolated data were downloaded on a regular grid with a resolution of 0.125° (approximately 13.9 km). The interpolated product is produced by ECMWF's Meteorological Interpolation and Regridding (MIR) package (Maciel et al., 2017) and is available via ECMWF's download portal at: https://apps.ecmwf.int/datasets/data/era20c-daily/levtype=sfc/type=an/. Temporal resolution is 3-hourly."

Figure 1. A snapshot of wind speed (shading & arrow length) and direction (arrow orientation) and mean sea level pressure (isobars) from ERA-20C. The search area is shown in red. There is one wind arrow per grid cell, giving an indication of the spatial resolution of the forcing product. Note that this is an interpolation onto a 0.125° regular grid from the native grid output from ERA-20. It is made available for download at this resolution via ECMWF's website.

2. The wind observations by Hussey have been interpolated to hourly values (Sec. 2.2). Was the same interpolation also performed for the 3 hourly ERA data?

We thank the reviewer for picking this up. Both sets of trajectories (ERA and Hussey) were computed at both 1 and 3-hourly resolution, producing negligible differences. However, it's true that the temporal resolution of the manuscript results should be consistent. We have thus changed the Hussey trajectories to 3-hourly to match the ERA ones.

We have added the following to Section 2.3:

"These data were linearly interpolated to 3-hourly resolution to match the ERA-20C data (see Section 2.2)..."

Sec. 2.4:

1. The drift trajectories are calculated from the wind data. But which method is used for the calculation, e.g. a simple Euler forward time step or a more advanced integration/ interpolation method?

Two methods were tested. At each time step, the new position is computed from the start position by supplying the drift direction and distance to:

1. The Vincenty formula (Vincenty, 1975) via the *m\_fdist.m* function for MATLAB by Pawlowicz (2020)

2. A simple trigonometric approach via the *ptlatlon.m* function for MATLAB by (Marty, 2018). Specifically:

 $lat_{end} = \operatorname{asin}\left(\sin(lat_{start})\cos\left(\frac{drift\_dist}{R_{earth}}\right) + \cos(lat_{start})\sin\left(\frac{drift\_dist}{R_{earth}}\right)\cos\left(drift\_bearing\right)$

 $lon_{end} = lon_{start} + atan2(sin(drift_{bearing})sin(\frac{drift_{dist}}{R_{earth}})cos(lat_{start})cos(\frac{drift_{dist}}{R_{earth}}) - sin(lat_{start})sin(lat_{end})$

Both methods produce nearly identical trajectories.

The following text has been added to Section 2.4.2:

"For each 3-hourly time step, the future position of the virtual sea ice floe is predicted by applying the wind-driven drift distance and direction to the Vincenty formula (Vincenty, 1975), as implemented in MATLAB by (Pawlowicz, 2020)."

2. Which time step is used for integration?

1-hourly and 3-hourly were tested. For the ERA, the 1-hourly represents an interpolated step, as the native resolution is 3-hourly. For the Hussey trajectories, both the 1 and 3-hourly are interpolated steps, as they are daily (mostly noon) observations. Since the trajectories were insensitive to the choice of time step, all trajectories presented in the manuscript utilised a 3-hourly time step.

The following added to Section 2.3:

"These data were linearly interpolated to 3-hourly resolution to match the ERA-20C data (see Section 2.2)..."

The following text has been added to Section 2.4.2:

"For each 3-hourly time step, the future position of the virtual sea ice floe is predicted by applying the wind-driven drift distance and direction to the Vincenty formula (Vincenty, 1975), as implemented in MATLAB by (Pawlowicz, 2020)."

Sec. 3.2: Positions of the Endurance were predicted for the entire period. Please provide the beginning and end of this period.

The period for which the drift simulation error was assessed was 18th January 1915 until 21st November 1915. This is the period from when Endurance became beset until the time of the sinking. The text shall be amended to include this information.

The following has been added to Section 3.1:

"Positions predicted by applying ERA-20C near-surface winds to virtual ice floes were reconstructed for the period 18 January 1915 until 21 November 1915, during which Endurance was beset and drifting in the ice pack."

Sec. 3.3:

1. In this section the winds obtained by ERA-20C and Hussey are compared indirectly by comparing trajectories which are an integrated form of winds. But the reader has no idea about the original wind data. Hence it is suggested, that the authors provide figures showing time series of wind speed and wind direction from both datasets, which is much more instructive for the comparison.

We thank the reviewer for this valuable suggestion. We have added Figure 4, a timeseries comparison of the two wind datasets, as well as Section 3.2 for commentary:

**"3.2 Comparison of ERA-20C winds with Hussey observations**

Figure 2 (2 shows a comparison between Hussey's wind recordings and the ERA-20C wind data. Whilst there are broad similarities between the two datasets, there are differences in speed, direction, and timing which account for material differences in corresponding trajectories. Broadly, both datasets suggest strong north-component winds at the start of the target period, which weaken and veer to become light south-component winds and increase in strength slightly by the end of the period. Concerning changes in direction, however, Hussey observed an earlier and more gradual veering from northerly winds (to southerlies by the start of 20 November) than ERA, which suggests winds veered later and more suddenly to become south-south-easterly by mid-morning on 21 November. Thereafter, Hussey's recordings indicate winds remained roughly south-south westerly until the end of the period, with southerly and south-south-easterly variations for short periods. ERA winds remained more uniformly south-easterly until the end of the period. Concerning speeds, whilst both datasets agree on generally high speeds, followed by a decrease and then an increase, there are two principal discrepancies. The first is a significant difference between the mornings of 19 November and 20 November (up to 20 knots) due to Hussey's observation of a much faster speed drop following the strong northerlies (ERA winds stay stronger for longer and never drop quite as low as Hussey's recordings). The second is a significant discrepancy from the afternoon of 21 November until the end of the period. Whilst both datasets suggest winds of around 10 knots by the afternoon of 21, Hussey's observed gradual increase to the end of the period is preceded by an initial drop to below 5 knots. ERA does not produce this decrease, so whilst it shows a similar gradual increase through the end of the period, an discrepancy of 5-10 knots persists."

Figure 2 (2 in manuscript). Time series comparison of wind speeds (top panel) and directions (bottom panel) between recordings from Hussey and the ERA-20C product. For the Hussey wind speeds (since Hussey reported Beaufort indices), the solid (dotted) line indicates wind speeds corresponding to the upper (lower) Beaufort index bound. The dashed line shows the mean for that index.

2.When comparing wind data from different sources one has to make sure, that these are taken at the same height above ground or have been interpolated to the same height. At which heights are the winds from Hussey and ERA-20C taken?

This is a valuable comment. In the initial set of results, the 10 m winds from ERA-20C were used without correction (as were the Hussey observations). We have now interpolated both the ERA-20C winds and Hussey's observed winds to the 2 m level by means of a logarithmic wind profile correction (Manwell et al., 2009), prior to use in the drift calculations. The choice of 2 m is a best guess to try and capture the winds which the floes of variable thickness (and irregular ridges) might have been forced by.

The following has been added to Section 2.3:

"We extracted 10 m wind speeds and directions from the ERA-20C dataset (Poli et al., 2016), adjusted them to the 2 m vertical level by applying a logarithmic profile correction (Manwell et al., 2009), and used them as a proxy to reconstruct the ice drift trajectory according to the methodology in Section 2.4. Figure A1 (Appendix A) illustrates this process, showing the simulated trajectory overlaid on ERA-20C wind and mean sea level pressure fields. For comparability, the 2 m level was selected as a best guess for the level at which Hussey's recordings were made (see Section 2.3), as well as a representative wind condition as experienced by the sea ice floes."

**A note:**

Data from NOAA-20CR reanalysis have been used by Etling (2017, DOI: 10.1127/metz/2017/0853) for investigating the atmosperic conditions of another famous polar expedition, the balloon flight of Andre and his crew in order to reach the North Pole in 1897 (which failed)

**References**

Manwell, J., McGowan, J., and Rogers, A.: Wind energy explained: theory, design, and application, 2nd ed., John Wiley & Sons Ltd., Chichester, 677 pp., 2009.

Pawlowicz, R.: M\_Map: A mapping package for MATLAB, https://www.eoas.ubc.ca/~rich/map.html, 2020.

Marty, R.: ptlatlon, https://nl.mathworks.com/matlabcentral/fileexchange/67254-ptlatlon, 2018.

Vincenty, T.: Direct and Inverse Solutions of Geodesics on the Ellipsoid with Application of Nested Equations, 88–93 pp., 1975.

**Response to RC2**

The authors present work that describes how they have used reanalysis data to better understand the sea ice conditions and trajectory before the Endurance sank in the Weddell Sea in November 1915. They find that the trajectory using reanalysis data has a more accurate wreck site to the actual location as compared to trajectories using observational data from the Endurance crew. Overall, this study is novel in that it presents some new methods for Polar marine archeology and how to use new data. I have some moderate to minor concerns that I think should be addressed before this is accepted for publication, including addition of another figure.

The authors would like to thank the reviewer for this positive and constructive input. Amendments made in response have both strengthened the manuscript directly and prompted us to enhance other aspects of the work. As the corresponding author I would also like to apologise for the delayed response. The timing of the review coincided with an Antarctic voyage, as well as other end-of-year academic commitments, thereby delaying the submission of these responses.

Specific comments:

Introduction: It would be nice to include details about the 1914-1915 expedition for those readers
unfamiliar with it. For example, it's worth mentioning briefly that the crew all survived and was
rescued from Elephant Island and how that compares in location to the wreck. Additionally, it
would be useful to summarize other expeditions searching for the Endurance and how the
successful approach in 2022 was different from those ventures. Finally, I didn't really understand
how the sinking happened and it would be useful to clarify: was the vessel stuck in the ice and
drifted several days before sinking or did it sink immediately? Could this account for some of the
poor estimate in location from Worsley's location?

Section 1 has been sub-divided, and a brief but comprehensive outline of the Trans-Antarctic Expedition provided (Section 1.1). All suggested details and some others have been included. Other plans and expeditions to search for the ship have been summarised in Section 1.2.

• Line 56 – Please clarify in text (possibly in the introduction) what "Ocean Camp" is and how that differs from the location of the Endurance.

The following has been added to Section 1.1:

"After drifting aboard the beset Endurance, having planned to wait until it broke free, Shackleton ordered the vessel abandoned in late October of 1915 due to severe damage inflicted by the crushing sea ice (Shackleton, 1919). Then, having attempted to march westward toward the islands of the Antarctic Peninsula in search of supplies and shelter, the crew was halted just a short distance from the stricken Endurance by the challenging ice conditions. There, approximately 2.5-3 km from the wreck, they established Ocean Camp, where they would await an improvement in conditions. After drifting with the sea ice for 10 months, and 25 days after being abandoned by the crew, Endurance finally sank during the late afternoon of 21 November 1915. Shackleton initiated a second march in late December 1915 but was again foiled by the ice conditions. Thus, Patience Camp was established just a week later, some 12 km from Ocean Camp, where the crew remained until early April 1916 (Shackleton, 1919). Following the break-up of the floe on which they were camping, the crew launched Endurance's three lifeboats on 9 April, sailing to and making landfall on Elephant Island on 15 April 1916."

• Line 58 – Why did Worsley add further offset? Is this known? Is the star on Figure 1 showing Worsley's location of the sinking include the offset?

It is not known exactly why Worsley added this offset. It may be that Worsley computed applied some kind of integration of their position change based on his estimate of the sea ice drift, though this is not known with certainty (and also not backed up by Hussey's wind observations). Bergman & Stuart (2018) suggest that this was simply an offset added to the nearest dependable fix (22nd Nov.) available to Worsley. The text has been simplified to avoid introducing unnecessary confusion, and reference made to (Bergman & Stuart, 2018) who provide a detailed analysis of these fixes.

• Line 78 – Please give the horizontal resolution of the ERA20 data (1degree? Higher resolution?) as this is relevant for how well it can resolve sea level pressure fields and near surface winds.

The following has been added to Section 2.2:

"Spatial resolution is approximately 125 km on the native ERA-20C triangular grid (Poli et al., 2016). However, interpolated data were downloaded on a regular grid with a resolution of 0.125" (approximately 13.9 km). The interpolated product is produced by ECMWF's Meteorological Interpolation and Regridding (MIR) package (Maciel et al., 2017) and is available via ECMWF's download portal at: https://apps.ecmwf.int/datasets/data/era20c-daily/levtype=sfc/type=an/. Temporal resolution is 3-hourly."

Line 93 and conclusion – You should mention that free drift is reasonable over short time scales \*in the Antarctic\* only - Kwok et al. 2017 is a good source for this (which is already in your reference list). This is relevant because you can't make the same assumptions in the Arctic (e.g. to have found the Erebus and Terror from the Franklin Northwest Passage Expedition). This is relevant because one of your main conclusions is that marine archaelogy in sea ice covered oceans can benefit from drift data, but the hemisphere may affect this technique. If you know of other Antarctic vessels that might benefit from this technique it would be useful to list them in the introduction or conclusion.

We thank the reviewer for this comment as the assumption of free-drift, and the tuning of freedrift parameters is certainly a key issue which materially affects the outcome of this study (and not only the general application of this method to marine archaeology). It prompted us to enhance our approach as explained below.

Section 2.4 has been split into sub-sections. Section 2.4.1 now describes, in detail, the merits, assumptions and implications of the free-drift approach, and a review of free-drift parameter. The text draws on the findings of (Doble & Wadhams, 2006; Holland & Kwok, 2012; Kottmeier et al., 1992; Kwok et al., 2017; Lu et al., 2011; Nakayama et al., 2012; Nie et al., 2022; Uotila, 2001; Uotila et al., 2000; Vihma et al., 1996; Vihma & Launiainen, 1993; Wamser & Martinson, 1993; Womack et al., 2022).

A further reason for including this analysis is because after further review of the above literature, it became clear that whilst the free-drift assumption is reasonable for the domain in question, and that wind is the primary forcing, drift trajectories are particularly sensitive to free-drift parameter settings (wind drag and turning angle). In particular, it is apparent that parameter settings derived from buoy experiments in the literature vary widely in space, time and even from instrument to instrument in roughly the same area. As Kottmeier et al. (1992) put it: *"Parameters change both from one data set (buoy, period, region) to the other and within a certain data set."*

This prompted us to change our approach from simply reusing the free-drift configuration settings used onboard during the Endurance22 expedition, to a more considered, comprehensive approach whereby setting were tuned in a series of sensitivity analyses.

Results of sensitivity testing is shown in Figure 1 below. Given the expanded envelope of results, Section 3, Results and Discussion, is vastly enhanced. Rather than one presumed scenario, describe a range of possible outcomes using different settings in many different simulations. We also explain, explicitly, remaining sources of uncertainty, and draw inferences based on features which are consistent throughout the envelop of results. Section 3.5 accounts explicitly for discrepancies our simulated sinking locations and, in the case of the idealised case, discrepancies between the required parameter values and more typical values per the literature. It also explains the general challenge of this under-constrained problem, and the strategic use of assumptions to investigate it.